# Evolution of Potentially Actionable Genomic Alterations in Advanced Prostate Cancer: A Real-World Analysis of Serial Circulating Tumor DNA Testing

**DOI:** 10.3390/cancers17183048

**Published:** 2025-09-18

**Authors:** Miguel Muniz, L. Jill Tsai, Jacob J. Orme, Leslie A. Bucheit, Spyridon P. Basourakos, Nancy Wei, Regina M. Koch, Zachary Scharf, Sounak Gupta, Adam M. Kase, Rodrigo Rodrigues Pessoa, Irbaz B. Riaz, Eugene D. Kwon, Jack R. Andrews, Daniel S. Childs

**Affiliations:** 1Department of Medical Oncology, Mayo Clinic, Rochester, MN 55905, USAchilds.daniel@mayo.edu (D.S.C.); 2Guardant Health, Palo Alto, CA 94304, USA; 3Department of Urology, Mayo Clinic, Rochester, MN 55905, USA; 4Department of Internal Medicine, Mayo Clinic, Rochester, MN 55905, USA; 5Department of Laboratory Medicine and Pathology, Mayo Clinic, Rochester, MN 55905, USA; 6Department of Medical Oncology, Mayo Clinic, Jacksonville, FL 32224, USA; 7Department of Urology, Moffitt Cancer Center, Tampa, FL 33612, USA; 8Division of Hematology and Medical Oncology, Mayo Clinic, Scottsdale, AZ 85259, USA; 9Department of Urology, Mayo Clinic, Phoenix, AZ 85054, USA

**Keywords:** prostate cancer, circulating tumor DNA, liquid biopsy, genomic testing

## Abstract

This study evaluated 479 advanced prostate cancer patients who had a baseline and at least one subsequent circulating tumor DNA (ctDNA) test performed. New and potentially actionable genomic alterations emerged in more than half of patients, revealing possible targets for on-label therapy, off-label therapy, and clinical trials. These findings underscore the potential of serial ctDNA profiling to guide personalized therapy and clinical trial selection by offering dynamic insights into genomic evolution.

## 1. Introduction

Prostate cancer remains a leading cause of death, and the number of advanced cases has almost doubled over the past decade [1]. Management of metastatic castrate-resistant prostate cancer (mCRPC) is nuanced, owing to disease heterogeneity and the growing number of treatment options. Increasingly, molecular profiling is used to inform therapeutic decisions, with select patients identified as candidates for immunotherapy or targeted treatments, such as poly (ADP-ribose) polymerases inhibitors (PARPi), based on the results of germline and somatic testing.

Traditionally, next generation sequencing has been performed on tumor tissue from the primary site of disease or a single metastatic lesion [2]. However, a single tissue-based biopsy provides limited data on spatial and temporal disease heterogeneity [3]. Due to the invasive nature of tissue biopsies, it is impractical to obtain biopsies of multiple metastatic sites at serial timepoints. Plasma-based circulating tumor DNA (ctDNA) assays, also known as “liquid biopsies,” require less-invasive peripheral blood draws. Active tumor shed can be analyzed via blood-based testing and provides dynamic disease data [4]. In this way, liquid biopsies can be used to serially capture a “real-time” molecular profile through the course of treatment. One such commercially available platform is Guardant360^®^, a plasma-based liquid biopsy approved for many solid-organ malignancies.

Some clinics reassess ctDNA at critical timepoints, such as diagnosis and each progression event, in hopes of identifying additional therapeutic and/or trial options. For men with advanced prostate cancer, changes in the tumor mutational profile occurring as a result of time and treatment pressures have been poorly characterized. Herein, we explore how often new and potentially actionable genomic alterations emerge on subsequent liquid biopsies (i.e., detection of new alterations that were not present on the initial ctDNA assessment). For those patients with potentially targetable alterations, we describe response rates to PARPi and immunotherapy.

## 2. Methods

### 2.1. Overall Cohort

We conducted a retrospective analysis of the Guardant360 database (Guardant Health, Inc., Redwood City, CA, USA) for patients with advanced prostate cancer who underwent ctDNA testing using the Guardant360 platform in the United States. At the time of patient testing and this analysis, Guardant360 assessed alterations in up to 83 genes and included reporting on blood-based tumor mutational burden (TMB) and microsatellite instability (MSI) (Appendix A). Deidentified data from Guardant Health used for research purposes is approved by the Advarra Institutional Review Board (Pro00034566) and complies with the United States Health Insurance Portability and Accountability Act regarding the determination and documentation of statistically deidentified data. As the data have been deidentified, a waiver of informed consent was approved in the Institutional Review Board (IRB) protocol.

In this study, we assessed patients with advanced prostate cancer and ctDNA detected between October 2020 and March 2023 (“Overall Cohort”) (Figure 1 [consort diagram]). Patients were filtered to identify those with multiple ctDNA tests (i.e., ≥2 tests) that reported ≥1 alteration within the testing period (*n* = 1441 patients). We performed the serial analysis only on the patients who had their first and second serial Guardant360 tests conducted within the study period (*n* = 479). Thus, although some patients had 2 or more samples within the testing period, timepoints prior to the study period were excluded and only two timepoints were assessed for the study.

Patients were assessed for actionable alterations, including primary and expanded actionability. Primary actionability was defined as alterations associated with a therapy approved in prostate cancer (Appendix A). Expanded actionability was defined as alterations associated with a therapy available in another cancer type or clinical trial availability. Primary and expanded actionability were examined for each patient’s first test (“Test 1 findings”) and for alterations found only on a subsequent test that was not detected on a patient’s first test (“Test 2 new findings”). With regard to PARPi therapy, at the time of patient testing and study analysis, Guardant360 assessed *BRCA1/2*, *ATM*, *CDK12*, *CHEK2*, *PALB2*, and *RAD51D* alterations.

We further assessed patients with TMB scores reported in two serial tests within the study period and analyzed patterns in TMB scores between serial tests as increasing (Δ > 0), decreasing (Δ < 0), or stable/no change (Δ = 0). The algorithm used to quantify TMB reports on 1 Mb of coding regions has been previously reported [5]. Prior studies in non-small cell lung cancer have demonstrated a positive association between immunotherapy outcomes and samples with a TMB score greater than the 80th percentile for a given cancer [5,6]. In a previous presentation, TMB scores reported on Guardant360 as used in this study were assessed by tumor type, and the 80th percentile of TMB scores in prostate cancer reported at 13.4 mut/Mb [7]. Thus, the threshold of high blood TMB (bTMB-H) was set at >13.4 mut/Mb for this analysis.

### 2.2. Mayo Clinic Cohort

The “Mayo Clinic Cohort” consists of patients in the “Overall Cohort” who received treatment for advanced prostate cancer at a Mayo Clinic site and provided authorization for their medical records to be used for research purposes. As such, this cohort provides more detailed clinical information—such as data on subsequent targeted treatments based on serial ctDNA results—which are not available for the larger Guardant cohort. This portion of the retrospective study was approved by the Mayo Clinic Rochester Institutional Review Board (study IRB#22-010856). Clinical characteristics were abstracted, including demographic data, clinical and pathologic stage, treatment history, and survival status. For those with treatment emergent genomic alterations, best biochemical (i.e., PSA) response to an “on-label” therapy (PARPi or immunotherapy) was recorded.

### 2.3. Statistical Analysis

The results were reported using descriptive statistics. These were summarized using medians and interquartile ranges for continuous variables and counts and proportions for categorical variables. Statistical analyses were performed using R version 4.5.1 (R Foundation for Statistical Computing, Vienna, Austria).

## 3. Results

### 3.1. Overall Cohort

#### 3.1.1. Frequency of Serial ctDNA Testing

A total of 1441 patients with advanced prostate cancer and serial testing were assessed. Most patients had 2 (*n* = 937, 65.0%), 3 (*n* = 252, 17.5%), or 4 tests (*n* = 127, 8.8%) during the study period (Figure 2). Among patients with multiple ctDNA tests, we focused our analysis on the subset of patients who had their first and a subsequent ctDNA test performed during the study period (*n* = 479). For this group of patients, the median interval between the first and second evaluable tests was 207 days (interquartile range [IQR], 114–346).

#### 3.1.2. Actionability of Results from Serial ctDNA Testing

Primary and expanded actionability from first and second ctDNA tests are shown in Figure 3. Test 1 identified potentially actionable data in 96.7% (*n* = 463) of patients. This included identification of genomic alterations with on-label implications for 45.9% (*n* = 220) of patients. Of note, 25% (*n* = 55) of these patients had an *AR* alteration, while the remaining 75% (*n* = 165) potentially qualified for PARPi based on alterations in genes included on a PARPi label, acknowledging some alterations likely associated with clonal hematopoiesis are included in these counts. Forms of other actionable data included identification of “off-label” therapy options for 40.5% (*n* = 194) and clinical trial availability for 96% (*n* = 460). It is worth noting that the results from a single ctDNA test commonly identified multiple forms of actionable information; for instance, a patient may have qualified for both an approved therapy and a clinical trial based on findings from the same test.

New, actionable data was identified on a subsequent ctDNA test (i.e., not present on first test [“Test 2 New Findings”]) for 57.2% (*n* = 274) of patients. Actionable data in the subsequent test had potential “on-label” implications for 16.7% (*n* = 80), including 72 patients (90%) with newly detected homologous recombination repair (HRR) alteration, 6 patients (7.5%) with acquisition of *AR* alterations, and 2 patients (2.5%) with *NTRK* alterations. Other expanded actionability included approved therapy options in other cancer types for 16.5% (*n* = 79) and clinical trial availability for 55.7% (*n* = 267).

#### 3.1.3. Acquisition of Alterations Impacting Homologous Recombination Repair (HRR) Genes

There were 232 patients with a detected HRR alteration in either their first or second tests (Figure 4A). The most common alterations were found in the *ATM* (20.3%, *n* = 97) and *CHEK2* (20%, *n* = 96) genes. *BRCA1* alterations were found in 4.4% of patients (*n* = 21) with 38% (*n* = 8) of alterations detected in the first test and 62% (*n* = 13) newly detected in the subsequent test. *BRCA2* alterations were found in 18% of patients (*n* = 86) with 59% (*n* = 51) detected in the first test and 41% (*n* = 35) newly detected in the subsequent test. *PALB2* alterations were found in 4% of patients (*n* = 19) with 47% (*n* = 9) detected in the first test and 53% (*n* = 10) newly detected in the subsequent test. *CDK12* and *RAD51D* alterations were found in 6.9% (*n* = 33) and 3.5% (*n* = 17) of patients, respectively. Figure 4B delineates the class or variety of each alteration within the HRR genes. The median (IQR) variant allele fraction (VAF) for HRR alterations detected on test 1 and 2 were 1% (0.3–15.9) and 0.3% (0.2–0.6), respectively.

#### 3.1.4. Changes in Tumor Mutational Burden

For patients with two or more TMB values available (*n* = 546, 9%), we describe the trend in TMB from the first to the most recent test (Table 1). The median time between tests was 141 days, and there was a median overall change of +0.72 mut/Mb. In total, 53% of patients had an increasing TMB score (median, +2.87 mut/Mb), 40% had a decreasing TMB score (median, −2.38 mut/Mb), and 7% of patients had no TMB change. Of those with an increasing TMB, 62 (21%) of patients had their TMB rise to meet a bTMB-high threshold of 13.4 mut/Mb, with a median (IQR) of 15.3 mut/Mb. None of these 62 patients had microsatellite instability or alterations in assessed mismatch repair deficiency genes on the subsequent test.

### 3.2. Outcomes for the Mayo Clinic Cohort

A total of 77 patients from the “Overall Cohort” received care at a Mayo Clinic site and developed treatment-emergent alterations with an associated “primary actionability” designation. For 49 (63.6%) patients, the acquired alteration was identified in an HRR gene, and 10 (12.9%) patients subsequently received a PARPi (Table 2). At the time of starting olaparib, all patients had castration resistant disease, and the median age was 62.5 years. PARPi was prescribed as third (*n* = 1), fourth (*n* = 2), or greater (*n* = 7) line therapy. Most patients (7/10) were prescribed a PARPi for copy number loss in an HRR gene, inclusive of single copy deletions and biallelic losses (Table 2). Amongst the other 3 patients, one had a frameshift mutation of *CHEK2* (0.3% cfDNA), one had an *ATM* splice site alteration (1% cfDNA), and one had a missense mutation in *CHEK2* (0.5% cfDNA). We evaluated the efficacy of olaparib in this cohort of patients. No PSA responses (0/10) were observed; all patients had rising PSA as the best biochemical response (Table 2). The median (IQR) duration of therapy was 7.9 (6.7–11.1) weeks. Only two patients received treatment for longer than 3 months (Patient 4 for 16 weeks and Patient 9 for 61 weeks).

Within this Mayo Clinic cohort, 28 patients developed a rise in TMB, which transitioned them from a blood-based “TMB-low” (<13.4 mut/Mb) to “TMB-high” designation (>13.4 mut/Mb). None of these patients had microsatellite instability (MSI-high) detected nor pathogenic alterations in mismatch repair genes. Only 2 of the 28 patients were subsequently exposed to immunotherapy as fifth and sixth-line therapy. Patient 11 was on treatment for 6.4 weeks, and patient 12 was on therapy for 9.9 weeks. The best response was progressive disease in both cases.

## 4. Discussion

The current guidelines for advanced prostate cancer endorse genomic testing, as the results provide valuable prognostic insights and help determine eligibility for additional biomarker-driven treatment approaches [8,9]. The best practices regarding if, when, and how to repeat somatic profiling for patients with advanced prostate cancer are not yet defined. This study was conducted to better understand how often new and potentially actionable genomic alterations, such as those impacting HRR genes, are detected only on subsequent ctDNA testing (i.e., not on initial testing). This is important because variable ctDNA shedding, low ctDNA content, and the limited sensitivity of assays for detecting copy number losses, deletions, and large structural variants may all contribute to the absence of alterations on the initial blood-based assay [10,11]. Theoretically, serial ctDNA testing may overcome some of these factors, particularly as advanced disease becomes more active and may confer higher rates of cell death and/or tumor shedding for ctDNA detection. Here, we report that more than half of patients had new and potentially actionable data emerging on their subsequent next-generation sequencing (NGS) test.

Increasingly, it is recognized that advanced prostate cancer evolves under therapeutic pressure [12]. Among patients receiving novel hormonal therapies, iterative ctDNA testing has been used to chronicle treatment-associated changes in the androgen receptor (AR), including increases in *AR* copy number, mutational frequency, and truncating rearrangements of the *AR* ligand binding domain [13,14]. Such alterations are enriched in advanced disease yet rarely found in untreated prostate cancer, and these changes contribute to therapeutic resistance [15]. Clones with AR pathway amplifications or mutations often become predominant at progression on AR-targeted therapies, reflecting convergent evolution toward AR-driven resistance [12]. AR pathway inhibitors are associated with more pronounced shifts than taxane chemotherapy and enrichment for alterations within the PI3K-AKT pathway [16]. However, far less data has been published on the emergence of other, potentially “druggable” somatic alterations. Paired, same-patient, treatment-naïve, and mCRPC biopsies have suggested that alterations in tumor suppressor genes and the PI3K/AKT pathway increase over time, but alterations in the DNA damage repair (DDR) pathway appeared to be more truncal [14].

Notably, in this study, we found that over 15% of patients with serial testing that detected a somatic alteration may qualify for an additional on-label therapy based on the results of a subsequent ctDNA test. The subsequent test may detect potentially relevant alterations in genes such as *BRCA2*, *BRCA1*, *CDK12*, and *PALB2*, which have been associated with radiographic progression free and/or overall survival benefit with PARPi treatment [17,18,19,20,21]. However, caution is needed in interpreting these results. At the time of this analysis, most commercial assays were not routinely evaluating for interference from clonal hematopoiesis (CH) in HRR-related genes, which can be discerned using paired leukocyte DNA or other, often bioinformatics-driven, methodologies. Up to 10% of men with mCRPC may be incorrectly considered good candidates for PARPi treatment due to the presence of CH variants [22]. Thus, the evolution of liquid biopsy assays that continue to lower the limit of detection and/or distinguish variants such as CH is critical for accurate interpretation of ctDNA results.

Often, treatment emergent alterations were detected at a low VAF (<1%). Low-VAF alterations in ctDNA may be a result of biological factors like genetic heterogeneity, early subclonal evolution, and clonal hematopoiesis of indeterminate potential (CHIP), or assay-related factors [23]. Alterations that are detected at a low VAF may still be clinically important rather than mere noise. Mizuna et al. compared treatment response in men with HRR mutated prostate cancer and ctDNA fractions between 0.4 and 2% versus ≥2%, finding no difference between groups [24]. Similar observations have been made in lung cancer supporting the broader importance of low VAF findings [25,26]. In a real-world study of more than 3000 patients receiving a targeted treatment for lung cancer, progression free survival for patients with druggable VAF < 1% was similar to those with VAF ≥ 1% [26]. Low VAF alterations are not unique to liquid biopsies, even tissue-based testing may identify primary driver and therapy-associated resistance alterations at relatively low VAFs [27]. It is essential to consider clinical context, including whether the gene is typically involved in prostate cancer biology, as well as the limitations of the testing assay.

There is insufficient data in this study to comment on how effective PARPi are for treatment-emergent alterations in HRR-related genes. Only 10 patients in the Mayo Clinic cohort received a PARPi, and no responses were observed. However, 5 of the 10 patients were prescribed olaparib for pathogenic alterations in *CHEK2* or *ATM*, which may contribute to the lack of responses, given more contemporary data suggests less benefit in such patients [17,28]. Additionally, many patients were heavily pre-treated (3+ lines of therapy, including platinum) at the time of PARPi intiation which may impact therapeutic response.

A second, clinically relevant observation is that bTMB is dynamic, and a subset of patients (21%) transitioned from a bTMB-low to bTMB-high designation (using an 80th percentile blood-based cut-off of >13.4 mut/Mb). Pembrolizumab is an on-label treatment for TMB-H tumors, but prior work has suggested that men with TMB-H prostate cancer without co-occurring MSI-H or mismatch repair deficiency are unlikely to respond to immunotherapy [29,30]. In the current cohort, only two such patients at Mayo Clinic were treated with immunotherapy and neither responded; notably, these patients were also heavily pre-treated, similar to the PAPRi cohort.

Many questions remain, but it is evident the approach of serial ctDNA sampling in advanced prostate cancer may increase the number of potential treatment options for many patients. Our analysis focused only on new changes that emerged on a subsequent ctDNA assay and identified a clinical trial for over 50% of patients. There are broader efforts, such as MatchMiner and others, using informatics platforms and open-source data to connect patients to genotype-driven trials [31]. ctDNA is a high-fidelity option that provides updated information on driver alterations and resistant clones, which can be used to replace or complement single site tissue biopsies. When selecting personalized treatment approaches, it may be ideal to use current rather than archival mutational profiling, as older results may become less representative over time due to disease evolution [16].

When approved treatments and clinical trials are exhausted, off-label prescribing can be considered after careful discussion with the patient. Use of off-label drugs in oncology is a controversial topic, but more than three-quarters of oncologists report prescribing anti-cancer therapies outside of their approved clinical indications [32]. In-person and virtual molecular tumor boards have only bolstered this practice [33,34,35], and success stories of genotype-driven approaches have been reported in other challenging malignancies like cancer of unknown primary and heavily pre-treated colorectal cancer [36,37]. In the current analysis, a therapy approved in other indications was found on the subsequent testing for nearly 17% of the cohort. Serial ctDNA analyses are useful for identification of emerging, resistant clones and guiding efforts to repurpose drugs and/or expand clinical trial opportunities.

This study has several limitations that should be acknowledged. Our analyses utilized a clinically focused targeted sequencing panel encompassing a select number of genes. At the time of analysis, the panel did not include all HRR genes or alterations that would qualify a patient for treatment with a PARPi; however, it did include those most important for therapeutic decision making regarding the use of PARPi, such as *BRCA2*, *BRCA1*, and *PALB2*. Other potentially relevant genes and non-coding regions are not assessed with this assay. Additionally, we do not have longitudinal quantification of overall tumor fraction nor results of CHIP analyses. This impacts our ability to comment on subclonal passenger events and non-prostate-derived clones. Our results do reflect the data used in real-world practice, and this makes our findings more applicable to practicing clinicians who may not have access to the more comprehensive research assays.

PSA has recommended intervals for follow-up assessment, such as every 3 or 6 months; however, there is currently no clear guidance regarding the optimal timing for repeating liquid biopsies. While the study shows that new alterations emerge, we are unable to provide clear guidance on the optimal timing or frequency of serial ctDNA testing. This is an area of future research. At our institution, providers typically repeat liquid biopsy when clonal evolution is anticipated, for example, at the time of biochemical or radiographic progression. Results from these timepoints can assist in informing subsequent therapy and evaluating trial eligibility. However, in current clinical practice, repeat testing is unlikely to alter management for patients who are responding to treatment.

Additional limitations include the small sample size of the Mayo Cohort, making it challenging to draw conclusions on the efficacy of PARPi or immunotherapy in the setting of treatment-emergent alterations.

## 5. Conclusions

Routine serial ctDNA testing provides a timely assessment of genomic changes, enhancing the identification of potentially actionable therapeutic targets and supporting precision oncology strategies. The present study demonstrates the emergence of new alterations over time; however, further analyses are required to establish the optimal intervals for retesting and to determine how these findings should guide treatment decisions.

## Figures and Tables

**Figure 1 cancers-17-03048-f001:**
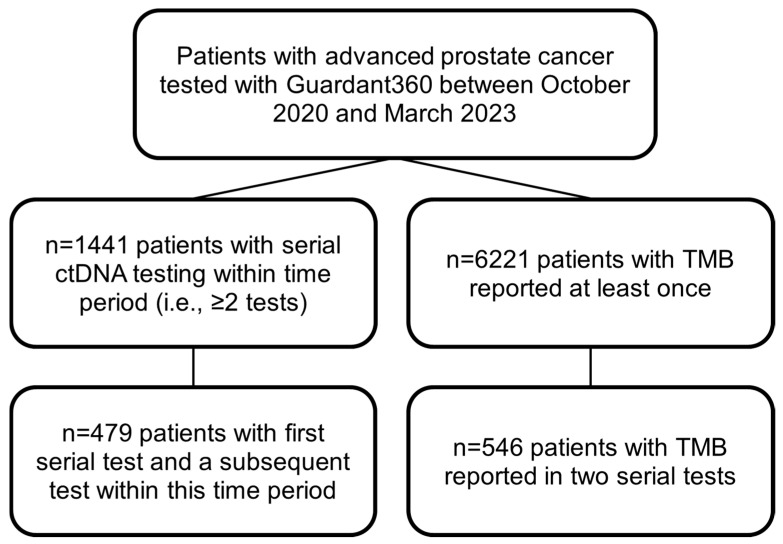
Consolidated Standards of Reporting Trials (CONSORT) flow diagram. Diagram depicts patients examined in study, delineated by those with serial alterations and TMB evaluated.

**Figure 2 cancers-17-03048-f002:**
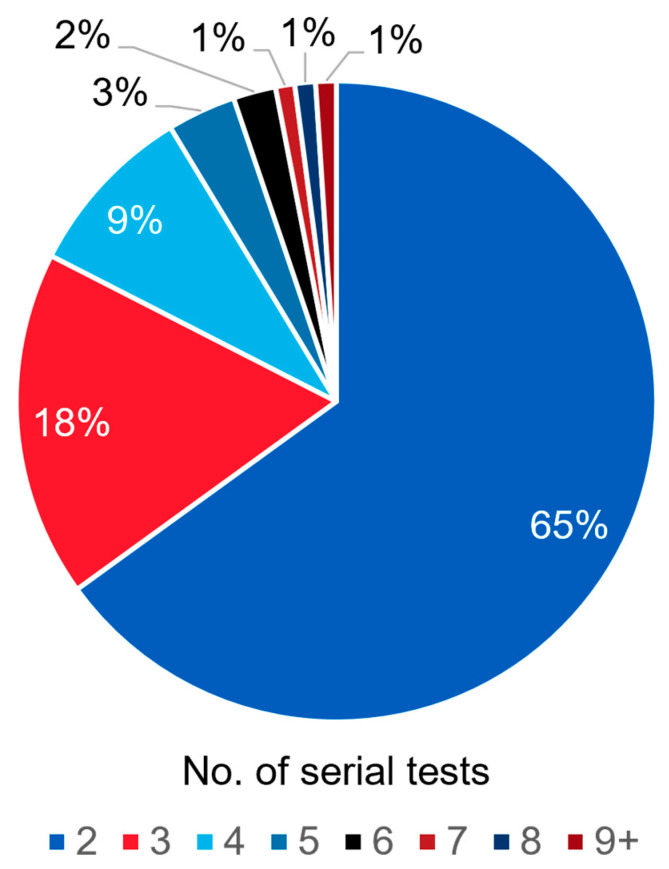
Pie chart depicting the frequency of patients with serial testing. Frequencies are delineated by number of serial tests.

**Figure 3 cancers-17-03048-f003:**
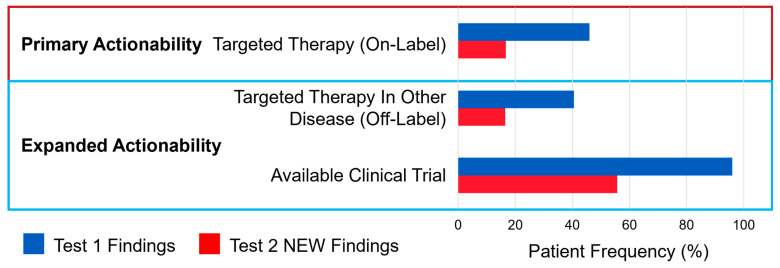
Patient frequencies with primary and expanded actionability. Test 1 findings refer to actionable alterations found on a patient’s first test, while Test 2 new findings refers to actionable alterations found on a patient’s second test that were not identified in the patient’s first test.

**Figure 4 cancers-17-03048-f004:**
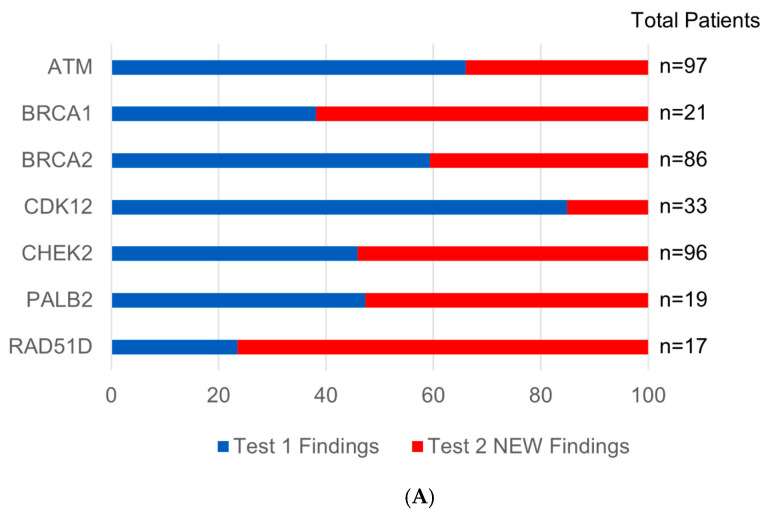
(**A**) Frequency of detected HRR alterations. Test 1 findings refer to actionable alterations found on a patient’s first test, while Test 2 new findings refers to actionable alterations found on a patient’s second test that were not identified in the patient’s first test. (**B**) Prevalence of HRR alterations delineated by class. Test 1 findings refer to actionable alterations found on a patient’s first test, while Test 2 new findings refer to actionable alterations found on a patient’s second test that were not identified in the patient’s first test. Copy number loss (CNL) includes single copy deletions and biallelic loss (single copy deletions + co-occurring loss of function [LOF] variant and homozygous deletions).

**Table 1 cancers-17-03048-t001:** TMB Dynamics and characteristics of patients with reported TMB scores in a first and subsequent next-generation sequencing (NGS) report.

Characteristic		
Median age, years	73	
Median Baseline bTMB, mut/Mb	8.61 (0.01–1164.75)	
Median time between serial tests in days	141	
**Pattern of bTMB Change over Time**	**Number of** **Patients, *n* (%)**	**Median Change in** **TMB (T^1^ to T^last^)**
Increasing	292 (53%)	+2.87 mut/Mb
Decreasing	216 (40%)	−2.38 mut/Mb
Stable/no change	38 (7%)	N/A
Total	546	+0.72 mut/Mb

Includes 62 (21%) patients with bTMB rising to meet bTMB-H threshold.

**Table 2 cancers-17-03048-t002:** Clinical, genomic and response characteristics of patients in the Mayo Clinic cohort receiving olaparib or pembrolizumab for treatment-emergent alteration.

Patient ID	Prior Lines of Therapy	Acquired ctDNA Alteration(s) Qualifying for Treatment (%cfDNA)	Drug	Baseline PSA, ng/mL	Best PSA Response, % of Baseline	Duration of Therapy, Weeks
1	Abiraterone, Enzalutamide, Docetaxel, Carboplatin	BRCA2 CNL CHEK2 CNL	Olaparib	185	+253	6
2	Abiraterone, Enzalutamide, Darolutamide, Docetaxel, Carboplatin	BRCA2 CNL CHEK2 CNL	Olaparib	61	+293	7
3	Abiraterone, Enzalutamide, Apalutamide, Docetaxel, Cabazitaxel, Carboplatin	CHEK2 D6fs (0.3%)	Olaparib	90	+218	11.1
4	Abiraterone, Enzalutamide, Darolutamide, Docetaxel, Carboplatin, LuPSMA	CHEK2 CNL	Olaparib	40	+115	16.4
5	Abiraterone, Enzalutamide, Apalutamide, Docetaxel, Cabazitaxel, Carboplatin, LuPSMA, Sipuleucel-T, Cisplatin + Etoposide	ATM SS SNV (1%)	Olaparib	36	+299	4
6	Abiraterone, Enzalutamide, Docetaxel, Carboplatin, LuPSMA, Ac-225	ATM CNL CHEK2 CNL	Olaparib	455	+154	11.1
7	Abiraterone, Enzalutamide, Docetaxel, Cabazitaxel, Carboplatin, LuPSMA	BRCA1 CNL CHEK2 CNL RAD51D CNL	Olaparib	101	+211	6.6
8	Abiraterone, Enzalutamide, Docetaxel, Cabazitaxel, Carboplatin, LuPSMA, Ac-225	CHEK2 D347A (0.5%)	Olaparib	98	+285	7.1
9	Abiraterone, Docetaxel	BRCA2 CNL	Olaparib	0.12	+192	61.1
10	Abiraterone, Docetaxel, LuPSMA	BRCA2 SCD	Olaparib	0.1	+100	8.6
11	Abiraterone, Apalutamide, Docetaxel, LuPSMA	TMB 15.3, MSS, pMMR	Pembrolizumab	121	+862	6.4
12	Abiraterone, Apalutamide, Docetaxel, Cabazitaxel, Carboplatin, LuPSMA	TMB 24.9, MSS, pMMR	Pembrolizumab	25	+141	9.9

## Data Availability

The data presented in this study are available on request from the corresponding author due to privacy and legal reasons.

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
