# Peer review of "Evolution of Potentially Actionable Genomic Alterations in Advanced Prostate Cancer: A Real-World Analysis of Serial Circulating Tumor DNA Testing"

_cancers, 2025, doi:10.3390/cancers17183048_

Round 1
Reviewer 1 Report
Comments and Suggestions for Authors
This manuscript presents a timely and clinically relevant real-world analysis of serial ctDNA testing in advanced prostate cancer, utilizing the Guardant360 assay. The study demonstrates that a significant proportion of patients develop new potentially actionable alterations over time, highlighting the dynamic nature of genomic evolution and the potential utility of repeated liquid biopsies. The work is well-structured, the methodology is sound, and the findings are of interest to the clinical and research communities in precision oncology. However, several points require clarification and further discussion to strengthen the manuscript.
- While the study shows that new alterations emerge, it does not provide clear guidance on the optimal timing or frequency of serial ctDNA testing. A brief discussion on whether testing should be triggered by progression, PSA rise, or at fixed intervals would enhance clinical relevance.
- The gene list is helpful. Please confirm that all genes listed were indeed included in the Guardant360 panel during the study period.
- The manuscript is generally well-written. Minor grammatical edits are recommended.
- Include a brief discussion on the clinical actionability of low-VAF alterations (e.g., median VAF of 0.3% in Test 2 HRR alterations). How should clinicians interpret such findings?
- The authors mentioned cancer therapy, the supramolecular method should be introduced. To support this claim, the following important studies should be cited: Coord. Chem. Rev. 2024, 517, 216054; Exploration 2023;20210111.
Author Response
Reviewer 1
This manuscript presents a timely and clinically relevant real-world analysis of serial ctDNA testing in advanced prostate cancer, utilizing the Guardant360 assay. The study demonstrates that a significant proportion of patients develop new potentially actionable alterations over time, highlighting the dynamic nature of genomic evolution and the potential utility of repeated liquid biopsies. The work is well-structured, the methodology is sound, and the findings are of interest to the clinical and research communities in precision oncology. However, several points require clarification and further discussion to strengthen the manuscript.
Thank you for these supportive comments and the suggestions for improvement.
Comment 1: While the study shows that new alterations emerge, it does not provide clear guidance on the optimal timing or frequency of serial ctDNA testing. A brief discussion on whether testing should be triggered by progression, PSA rise, or at fixed intervals would enhance clinical relevance.
Response 1: This is an important question, but unfortunately, our study was not designed to answer the question of optimal timing or frequency of serial testing. We have added this as a limitation and area of future study. “PSA has recommended intervals for follow-up assessment, such as every 3 or 6 months; however, there is currently no clear guidance regarding the optimal timing for repeating liquid biopsies. While the study shows that new alterations emerge, we are unable to provide clear guidance on the optimal timing or frequency of serial ctDNA testing. This is an area of future research. At our institution, providers typically repeat liquid biopsy when clonal evolution is anticipated, for example, at the time of biochemical or radiographic progression. Results from these timepoints can assist in informing subsequent therapy choices and evaluating trial eligibility. However, in current clinical practice, repeat testing is unlikely to alter management for patients who are responding to treatment.”
Comment 2: The gene list is helpful. Please confirm that all genes listed were indeed included in the Guardant360 panel during the study period.
Response 2: We double-checked with the Guardant team and confirmed that all the genes listed were included in the Guardant360 panel during the study period.
Comment 3: The manuscript is generally well-written. Minor grammatical edits are recommended.
Response 3: Thank you! We carefully reviewed grammar, clarity, and readability to the best of our ability. See tracked changes.
Comment 4: Include a brief discussion on the clinical actionability of low-VAF alterations (e.g., median VAF of 0.3% in Test 2 HRR alterations). How should clinicians interpret such findings?
Response 4: The data on this topic is emerging quickly, and we agree that additional discussion of the topic might be helpful to readers. We have added the following: “Often, treatment emergent alterations were detected at a low VAF (<1%). Low-VAF alterations in ctDNA may be a result of biological factors like genetic heterogeneity, early subclonal evolution, and clonal hematopoiesis of indeterminate potential (CHIP), or assay-related factors [21]. Alterations that are detected at a low VAF may still be clinically important rather than mere noise. Mizuna et al. compared treatment response in men with HRR mutated prostate cancer and ctDNA fractions between 0.4%–2% versus ≥2%, finding no difference between groups [22]. Similar observations have been made in lung cancer supporting broader importance of low VAF findings [23,24]. In a real-world study of more than 3000 patients receiving a targeted treatment for lung cancer, progression free survival for patients with druggable VAF <1% was similar to those with VAF ≥1% [24]. Low VAF alterations are not unique to liquid biopsies, even tissue-based testing may identify primary driver and therapy-associated resistance alterations at relatively low VAFs [25]. It is essential to consider clinical context, including whether the gene is typically involved in prostate cancer biology, as well as the limitations of the testing assay.”
Comment 5: The authors mentioned cancer therapy, the supramolecular method should be introduced. To support this claim, the following important studies should be cited: Coord. Chem. Rev. 2024, 517, 216054; Exploration 2023;20210111.
Response 5: Thank you for this suggestion. This is an interesting topic that should be explored in greater depth in future papers.
Reviewer 2 Report
Comments and Suggestions for Authors
Muniz and colleagues investigate 479 advanced prostate cancer patients for tDNA tests. The hypothesis is of some interest but no clinically actionable results are presented beyond general comments.
Comments include:
*Why is the Mayo short treated differently as the overall cohort? What are the effects on data, conclusions etc.?
*Abbreviations are common and often unexplained, starting with ctDNA in the title and Simple Summary. A list of abbreviations might also be a good idea.
*The discussion of limitations under Discussion is essential but needs to permeate through to the Abstract, Conclusions etc.
Author Response
Reviewer 2
Muniz and colleagues investigate 479 advanced prostate cancer patients for tDNA tests. The hypothesis is of some interest but no clinically actionable results are presented beyond general comments.
Comment 1: Why is the Mayo short treated differently as the overall cohort? What are the effects on data, conclusions etc.?
Response 1: The Mayo Clinic cohort represents a more limited subset, which is why its findings are less emphasized in the results and discussion. However, this cohort provides more detailed clinical information—such as data on subsequent treatments based on serial ctDNA and best response—which is not available for the larger Guardant cohort. The study was designed with the intent to analyze both the overall Guardant database and the Mayo subset. Although the number of Mayo Clinic patients receiving genomics-guided therapy was smaller than anticipated, the findings were still included in accordance with the original analytic plan. We have added additional context to the Methods section to explain this rationale.
Comment 2: Abbreviations are common and often unexplained, starting with ctDNA in the title and Simple Summary. A list of abbreviations might also be a good idea.
Response 2: This is an excellent suggestion, thank you. We have introduced each abbreviation at first mention and added a list of abbreviations at the end of the manuscript.
Comment 3: The discussion of limitations under Discussion is essential but needs to permeate through to the Abstract, Conclusions etc.
Response 3: Thank you for this suggestion. We have added comments on limitations to the abstract and conclusions to emphasize the importance of further research.